# Outcomes and Prognostic Factors in Critical Patients with Hematologic Malignancies

**DOI:** 10.3390/jcm12030958

**Published:** 2023-01-26

**Authors:** Chieh-Lung Chen, Sing-Ting Wang, Wen-Chien Cheng, Biing-Ru Wu, Wei-Chih Liao, Wu-Huei Hsu

**Affiliations:** 1Division of Pulmonary and Critical Care Medicine, Department of Internal Medicine, China Medical University Hospital, Taichung 404, Taiwan; 2Division of Hematology and Oncology, Department of Internal Medicine, China Medical University Hospital, Taichung 404, Taiwan; 3School of Medicine, China Medical University, Taichung 404, Taiwan; 4Department of Life Science, National Chung Hsing University, Taichung 402, Taiwan; 5Ph.D. Program in Translational Medicine, National Chung Hsing University, Taichung 402, Taiwan; 6Rong Hsing Research Center for Translational Medicine, National Chung Hsing University, Taichung 402, Taiwan; 7Department of Respiratory Therapy, China Medical University Hospital, Taichung 404, Taiwan; 8Center for Hyperbaric Oxygenation Therapy, China Medical University Hospital, Taichung 404, Taiwan; 9Critical Medical Center, China Medical University Hospital, Taichung 404, Taiwan

**Keywords:** hematologic malignancies, intensive care unit, prognosis

## Abstract

Patients with hematologic malignancies (HMs) have a significantly elevated risk of mortality compared to other cancer patients treated in the intensive care unit (ICU). The prognostic impact of numerous poor outcome indicators has changed, and research has yielded conflicting results. This study aims to determine the ICU and hospital outcomes and risk factors that predict the prognosis of critically ill patients with HMs. In this retrospective study, conducted at a referral hospital in Taiwan, 213 adult patients with HMs who were admitted to the medical ICU were evaluated. We collected clinical data upon hospital and ICU admission. Using a multivariate regression analysis, the predictors of ICU and hospital mortality were assessed. Then, a scoring system (Hospital outcome of critically ill patients with Hematological Malignancies (HHM)) was built to predict hospital outcomes. Most HMs (76.1%) were classified as high grade, and more than one-third of patients experienced a relapsed or refractory disease. The ICU and hospital mortality rates were 55.9% and 71.8%, respectively. Moreover, the disease severity was high (median Sequential Organ Failure Assessment (SOFA) score: 11 and Acute Physiology and Chronic Health Evaluation (APACHE II) score: 28). The multivariate analysis revealed that high-grade HMs, invasive mechanical ventilation requirement, renal replacement therapy initiation in the ICU, and a high SOFA score correlated with ICU mortality. Furthermore, a higher HHM score predicted hospital mortality. This study demonstrates that ICU mortality primarily correlates with the severity of organ dysfunction, whereas the disease status markedly influences hospital outcomes. Furthermore, the HHM score significantly predicts hospital mortality.

## 1. Introduction

Over the last several decades, cancer survival rates have been consistently improving, resulting in an increasing number of patients who require intensive care unit (ICU) treatment for critical illnesses that are either directly or indirectly caused by cancer [1]. Unlike other ICU patients with cancer, patients with hematologic malignancies (HMs) are usually more ill and have a higher mortality rate [2,3]. Developments in diagnosis, chemotherapy and conditioning regimens, innovative treatment regimens, and intensive care have resulted in better outcomes in patients with HMs [4,5,6]. Therefore, physicians now tend to admit patients with HMs to the ICU [7,8].

Some previously identified poor outcome indicators include older age [9], neutropenia [10,11,12], disease progression [13], allogeneic hematopoietic stem cell transplantation (HSCT) [14,15], graft versus host disease (GVHD) [16], high Sequential Organ Failure Assessment (SOFA) score [17], high Simplified Acute Physiology Score (SAPS) II [12,18,19], high Acute Physiology and Chronic Health Evaluation (APACHE) II score [9,10,12,20], invasive mechanical ventilation (IMV) [10,15,17,21], renal replacement therapy (RRT) [19,20], hepatic dysfunction [18,20], and invasive fungal infection [14]. Nevertheless, the prognostic impact of some variables has changed, and the related research has yielded conflicting results [22]. Besides, limited studies have explored this patient group in the Asian population.

With substantial progress in supportive care and cancer treatment in recent years, this study aims to examine the outcomes of patients with HMs admitted to the ICU in Taiwan and determine the predictors of ICU and hospital mortality.

## 2. Materials and Methods

### 2.1. Study Design and Patients

This retrospective single-center observational study was conducted in a tertiary referral hospital in mid-Taiwan, a 1722-bed hospital with a 45-bed adult medical ICU and a 7-bed bone marrow transplantation facility. Around 80 HSCT procedures are performed annually in the hospital.

The institutional review board of the China Medical University Hospital approved this study (CMUH109-REC3-137). The requirement for informed consent was omitted as the study design was retrospective and no personally identifiable information was collected. Data were collected for all adult (aged ≥18 years) patients admitted to the ICU with HMs as the primary diagnosis or concurrent comorbidity over 44 months (1 January 2017–31 August 2020). We reviewed the hospital information system and medical records. Of note, patients admitted to the ICU for surgical interventions or postoperative monitoring were excluded. For patients admitted to the ICU more than once, only the first admission was included (Figure 1). Furthermore, patients who died within 24 h of ICU admission were excluded from the final analysis. In our hospital, the decision to admit a patient with HMs to the ICU from the emergency department is made by an emergency physician and intensivist, considering factors such as organ dysfunction and the need for advanced life support (IMV, RRT, or continuous RRT). For patients who have already been admitted to our hospital, the decision is made by the primary care hematologist.

### 2.2. Data Collection

The variables recorded included patient characteristics, type of HM, disease status during admission, previous HSCT, the presence of GVHD, severe neutropenia, the reason for ICU admission, the need for IMV, and RRT initiation in the ICU. We assessed the severity of illness at ICU admission using the SOFA score, SAPS II, and APACHE II score. All the scores were calculated using the patient data collected during ICU admission or within 24 h of ICU admission.

### 2.3. Definitions

In this study, HMs were classified into the following categories: acute myeloid leukemia (AML), acute lymphoblastic leukemia (ALL), chronic lymphoblastic leukemia, chronic myeloid leukemia, Hodgkin’s lymphoma, aggressive non-Hodgkin’s lymphoma (NHL), indolent NHL, and multiple myeloma. Of these, AML, ALL, and aggressive NHL were classified as high-grade [3].

The disease status was defined as complete remission, partial remission, stable disease, and relapsed or refractory disease. In patients with newly diagnosed HMs without treatment or unexpected hospital admission before the assessment of tumor response to cancer therapeutics, the disease status was labeled as “non-evaluable.” Severe neutropenia was defined as an absolute neutrophil count < 500/mm^3^, and severe neutropenia for >7 days was defined as prolonged neutropenia. An ICU-acquired infection was defined as a new infection that developed at least 48 h after ICU admission.

ICU mortality was defined as the number of patients with HMs who died in the ICU from any cause. This included deaths due to the progression of the underlying HMs, complications from treatment, or any other cause that occurred while the patient was in the ICU. Hospital mortality was determined in a similar manner.

### 2.4. Statistical Analyses

All statistical analyses were performed using SPSS ver. 25 (SPSS Inc., Chicago, IL, USA). For continuous data that were normally distributed, a *t*-test was used for analysis and the results were presented as mean ± standard deviation (SD). For ordinal and non-normally distributed data, the median and interquartile range (IQR) were presented and the differences between groups were determined using the Mann–Whitney U test. In contrast, the categorical variables were presented as numbers and percentages and analyzed using a chi-square test. Using a univariate analysis, we calculated the odds ratio (OR) of mortality, and the statistically significant variables were used in the multivariate logistic regression model analysis. Besides, the strength of association was presented as an OR and 95% confidence interval (CI). In this study, all the tests were two-sided, and we considered *p* < 0.05 as statistically significant.

## 3. Results

During the study period, 4767 patients were admitted to the medical ICU, of which 1239 had an underlying or current diagnosis of any type of malignancy, including 277 patients with HM. After excluding the patients admitted for surgical indications, postoperative monitoring, readmission to the ICU, and who died within 24 h of ICU admission, we examined 213 patients, representing 4.5% of all ICU admissions.

### 3.1. Patient Characteristics

The patients’ mean (SD) age was 59.9 (15.2) years, and 37.1% were females (Table 1). All the patients had a confirmed HM based on the pathological examination results or medical records, including 44.1% with leukemia, 43.7% with lymphoma, and 12.2% with plasma cell dysplasia. In addition, 76% of the HMs were classified as high-grade. Nearly 20% of the patients received allogeneic HSCT, and 6.1% underwent autogenic HSCT, of which 12.7% had GVHD. Furthermore, 71 patients (33.3%) had a relapsed or refractory disease. A total of 30 patients (14.1%) had a newly diagnosed HM in the ICU.

The leading reasons for ICU admission were pneumonia (45.1%) and septic shock (25.8%). Most patients (92.5%) needed IMV, and 35.2% of the patients required RRT initiation in the ICU. The median (IQR) SOFA score was 11 (9–15); SAPS II, 63 (51.5–77); and APACHE II score, 28 (23–34), as shown in Appendix A.

### 3.2. Outcomes

The ICU and hospital mortality rates were 55.9% and 71.8%, respectively. The causes of death in the hospital are shown in Appendix A. The comparisons between survivors and non-survivors with HMs are illustrated in Table 2. The multivariate logistic regression analysis established that the diagnosis of high-grade HMs (OR, 2.70; 95% CI: 1.20–6.07), IMV requirement (OR, 7.19; 95% CI: 1.50–34.42), RRT initiation in the ICU (OR, 5.43; 95% CI: 2.51–11.72), and SOFA score (OR, 1.16; 95% CI: 1.06–1.28) independently correlated with ICU mortality (Table 3). Of the three commonly used disease severity scores in the ICU, we chose the SOFA score for the logistic regression analysis to avoid collinearity and for practical purposes.

The factors independently correlating with hospital mortality comprised the following: relapsed or refractory disease (OR, 6.35; 95% CI: 2.21–18.29), non-evaluable disease status (OR, 2.78; 95% CI: 1.05–7.36), IMV (OR, 4.00; 95% CI: 1.03–15.50), RRT initiation in the ICU (OR, 6.01; 95% CI: 2.09–17.28), SOFA score ≥ 11 (OR, 2.66; 95% CI: 1.23–5.76), and ICU-acquired infection (OR, 2.42; 95% CI: 1.08–5.43), as shown in Table 4. For practical purposes, we dichotomized the SOFA score into two groups, ≥11 points and <11 points, based on data that had a maximal Youden’s index.

### 3.3. Construction of the Scoring System

In this study, the construction of a scoring system, the ‘Hospital outcome of critically ill patient with Hematological Malignancies’ (HHM) score, was based on the results of logistic regression assessing predictors for hospital mortality: relapsed or refractory disease, non-evaluable disease status, IMV requirement, RRT initiation in the ICU, SOFA score ≥ 11, and ICU-acquired infection. The risk index was assigned weights using a regression coefficient-based scoring system [23], where the β coefficients were rounded to the nearest integer by dividing by the smallest coefficient and then added together to calculate the HHM scores (Table 5). The area under the receiver–operating characteristic (AUROC) curve to predict hospital mortality was 0.81 (95% CI: 0.75–0.87; Appendix A). Furthermore, an HHM score of ≥5 resulted in >80% hospital mortality in our cohort (Table 5).

## 4. Discussion

This study is among the most comprehensive, single-center studies of critically ill patients with HMs, with a higher disease severity compared to previous studies (as shown in Appendix A), to our knowledge. Moreover, this is the first study in Taiwan to report the prognostic factors in this patient group. We re-evaluated the prognostic significance of previously published factors. The findings reveal that ICU mortality primarily correlates with the severity of organ dysfunction, while the disease status markedly influences the hospital outcomes. Furthermore, a robust predictive model is constructed for hospital mortality, specifically for critically ill patients with HMs.

Previous research has identified a number of prognostic factors for critically ill patients with HMs, including the presence of neutropenia, a poor performance status, advanced disease status, the use of IMV, the use of vasopressors, the need for RRT, and a history of HSCT. While scoring systems such as APACHE II and SAPS II were reported extensively, the SOFA score was less validated in predicting the ICU outcome of critically ill patients with HMs. In this study, the severity of organ dysfunction represented by high SOFA scores, and the need for IMV or RRT independently correlated with ICU mortality. Perhaps, ’traditional’ risk factors, such as transplantation status and neutropenia, are no longer predictive [24]. Our findings corroborate previous studies, in that the presence of organ failure was the primary factor in poor ICU outcomes [10,14,17,20,24,25]. We chose the SOFA score for the multivariate analysis due to its practicality in clinical practice and its previous use in determining the duration of time-limited trials for cancer patients in a critical condition [26]. Early ICU admission of patients at risk of multiorgan failure had been recommended [4], and early intervention before ICU admission in critically ill patients with cancer has been proven to decrease in-hospital mortality [27]. Overall, we suggest timely admission to the ICU in patients with a high risk of severe organ dysfunction [4,28,29]. Notably, underlying HM disease, neutropenia, and even post-HSCT status should not be considered a reason for delayed ICU admission.

In the regression analysis for hospital mortality, both high-grade HMs and disease status were found to be significantly related to hospital mortality. However, only the disease status was significant in the multivariate analysis. This can be explained by the fact that patients with disease progression or those just starting antineoplastic treatment tend to receive more intensive treatment after ICU discharge, which may be related to a longer hospital stay, an increased risk of treatment-related complications and subsequent infections.

A previous study reported that the ICU and hospital mortality rates in patients with HMs in the ICU were 24.8–66% and 43–77%, respectively [15]. In our study, the ICU and hospital mortalities were 55.9% and 71.8%. The difference in mortality rate can be attributed to differences in admission and discharge criteria, variations in disease severity, as well as the execution of end-of-life (EOL) decisions. Moreover, the disease severity at ICU admission was higher in our study than in previous studies (Appendix A). Besides, more than one-third of our patients experienced a relapsed or refractory disease; this phenomenon is possibly associated with the Taiwanese national health insurance system and its accessibility, short waiting times, universal insurance coverage, and low cost [30]. It has been reported that the use of aggressive care, including ICU admission, in the last month of life among cancer patients is more prevalent in Taiwan compared to other countries [31]. A national policy promoting hospice care led to a significant increase in hospice utilization and decreased the use of invasive EOL care [32]; however, determining the EOL phase for patients with HMs remains challenging [33]. While the cancer disease progression did not correlate with ICU mortality in our study, it independently correlated with hospital mortality, suggesting that a proportion of patients with HMs who survived from ICU eventually died in the hospital (Appendix A). Thus, the ICU admission policy for this patient group merits comprehensive discussion by hematologic specialists and intensivists, especially amid the ICU admission surge and scarcity of ICU beds witnessed during the COVID-19 crisis.

In addition, we identified ICU-acquired infections as a poor prognostic factor for hospital outcomes. The morbidity and mortality related to these infections are typically higher owing to the immunocompromised state of critically ill cancer patients. Reportedly, a high index of clinical suspicion is essential for early identification, isolation, and treatment [34]. A total of 82 patients were recorded as having ICU-acquired infections. Multidrug-resistant pathogens were detected in 59.8% (49 patients), while Stenotrophomonas maltophilia and Candidemia accounted for 17.1% (14 patients) and 8.5% (7 patients) of isolations, respectively (Appendix A). With the introduction of syndromic multiplex molecular tests for rapid diagnosis for pathogen- and drug-resistant genes [35,36], clinicians could optimize early antibiotic treatment for multi-drug resistant pathogens in patients at risk, further improving the outcomes among critically ill patients with HMs.

Despite the extensive examination of the severity scoring systems, such as APACHE II and SAPS II, in predicting ICU mortality, their prognostic value for hospital mortality remains less validated. Sculier et al. reported the AUROC for predicting the hospital outcomes in cancer as 0.67 and 0.60 for SAPS II and APACHE II, respectively, and concluded that, after recovering from the critical illness, the characteristics related to the neoplastic disease regain their independent impact on further survival [37]. Another study examined the effectiveness of six severity of illness scoring systems in patients with cancer needing ICU admission [38], including APACHE II, APACHE III, SAPS II, and the Cancer Mortality Model, and the AUROC for predicting hospital mortality was 0.754, 0.812, 0.815, and 0.795 respectively. However, in that study [38], the study population comprised both solid tumors and HMs, which we consider to be different entities. Furthermore, we constructed the HHM score to predict hospital morality among critically ill patients with HMs, with an AUROC of 0.811.

This study has several limitations worth acknowledging. First, being a retrospective single-center study, our results might not be applicable to other hospitals. However, our sample size was large, and we analyzed data collected over 44 months from one of the largest medical centers in Taiwan. Second, the mortality rate was higher in our cohort than in previous studies [15]. Nevertheless, the severity index was higher in our study, and comparing the crude mortality with those in other studies was challenging owing to the variations in the admission and discharge criteria. Besides, the disease severity was higher in our patients, and our results contribute to the current evidence regarding the outcomes of critically ill patients with HMs and high disease severity. Third, due to the retrospective study design and the lack of recorded performance status before hospital and ICU admission, we were unable to investigate the impact of performance status on the ICU or hospital outcomes. Fourth, although we found that the SOFA score at ICU admission was the independent factor for both ICU and hospital outcomes, we did not record the SOFA score during the later hospitalization. Thus, the prognostic value of the dynamic change of SOFA could not be stratified in our study. Fifth, due to the retrospective nature of the study, we did not have the ability to record the detailed and dynamic adjustments of vasopressor doses and the duration of renal replacement therapy, which may have had an impact on our patients’ outcomes. Sixth, we did not have information on the implementation and impact of end-of-life care. Finally, our sample size was insufficient to internally validate the outcome prediction model. Therefore, additional study is needed to thoroughly assess the external validity of the HHM score in a large, separate group of individuals. Moreover, future studies should focus on the long-term outcomes after ICU discharge, including quality of life, new epidemiology and challenges in ICU admission in the era of immunotherapy, and the impact of molecular tests for the rapid diagnosis of pathogen- and drug-resistant genes.

## 5. Conclusions

This study establishes that ICU mortality primarily correlates with the severity of organ dysfunction, whereas the disease status significantly influences the hospital outcomes. Furthermore, the HHM scoring system significantly predicts hospital mortality.

## Figures and Tables

**Figure 1 jcm-12-00958-f001:**
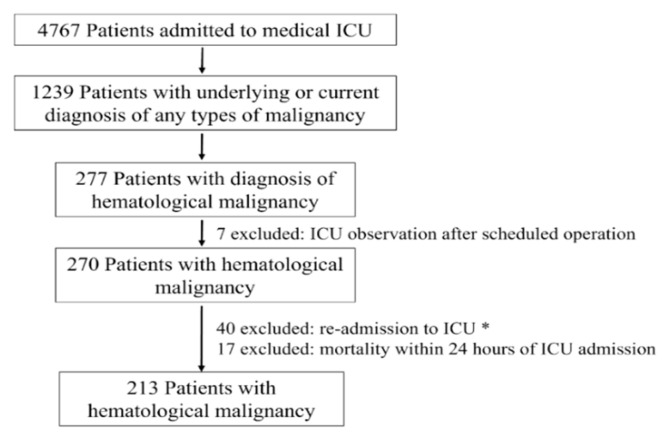
Study design and flow chart. ICU, intensive care unit. *** If the patient was admitted to the ICU more than once, the first admission was included.

**Table 1 jcm-12-00958-t001:** Patient characteristics.

	All (n = 213)
Age (years)	59.9 ± 15.2
Female	79 (37.1%)
BMI	22.6 ± 3.7
Cancer type	
Leukemia	94 (44.1%)
Lymphoma	93 (43.7%)
Plasma cell dysplasia	26 (12.2%)
High-grade hematological malignancies ^a^	162 (76.1%)
Hematopoietic stem cell transplantation	
Allogeneic	42 (19.7%)
Graft versus host disease	27 (12.7%)
Autogenic	13 (6.1%)
Disease status	
Complete remission	30 (14.1%)
Partial remission	11 (5.2%)
Stable disease	5 (2.3%)
Relapsed or refractory	71 (33.3%)
Non-evaluable	96 (45.1%)
Severe neutropenia ^b^	97 (45.5%)
Prolonged severe neutropenia ^c^	60 (28.2%)
Cancer diagnosed in ICU	30 (14.1%)
Main reason for ICU admission	
Pneumonia	96 (45.1%)
Septic shock	55 (25.8%)
Non-CNS bleeding	11 (5.2%)
Hematologic emergencies ^d^	11 (5.2%)
Neurological disorder ^e^	10 (4.7%)
Cardiac arrest	8 (3.8%)
Heart failure	5 (2.3%)
Metabolic disorder ^f^	3 (1.4%)
Renal failure	2 (0.9%)
Liver failure	2 (0.9%)
Large airway obstruction	2 (0.9%)
Others	8 (3.8%)
Severity of illness scores upon admission to the ICU	
SOFA score	11 (9–15)
SAPS II	63 (51.5–77)
APACHE II score	28 (23–34)
ICU intervention	
Invasive mechanical ventilation	197 (92.5%)
Renal replacement therapy initiation	75 (35.2%)
ECMO	4 (1.9%)
First line antineoplastic treatment in ICU	33 (15.5%)

^a^: Acute myeloid leukemia, acute lymphocytic leukemia, and aggressive non-Hodgkin’s lymphoma were classified as high-grade diseases (see text); ^b^: Absolute neutrophil count < 500/mm^3^; ^c^: Prolonged duration of severe neutropenia for >7 days. ^d^: Tumor lysis syndrome, differentiated syndrome, and malignancy related hypercalcemia; ^e^: Meningitis, malignancy with CNS involvement, suspect CNS immune-related adverse event, subdural hematoma, and subarachnoid hemorrhage; ^f^: Diabetic ketoacidosis and metformin related lactate acidosis. Abbreviations: BMI = body mass index; ICU = intensive care unit; CNS = central nervous system; SOFA = Sequential Organ Failure Assessment; SAPS = Simplified Acute Physiology Score; APACHE = Acute Physiology and Chronic Health Evaluation; ECMO = extracorporeal membrane oxygenation.

**Table 2 jcm-12-00958-t002:** Comparisons between survivors and non-survivors with hematological malignancies.

Characteristics	ICU Survivors (N = 94)	ICU Non-Survivors (N = 119)	*p* Value	Hospital Survivors (N = 60)	Hospital Non-Survivors (N = 153)	*p* Value
Age	60.0 ± 16.1	59.8 ± 14.5	0.93	59.4 ± 14.8	60.1 ± 15.4	0.772
Female	33 (35.1%)	46 (38.7%)	0.594	22 (36.7%)	57 (37.3%)	0.936
Body mass index	22.8 ± 3.9	22.5 ± 3.5	0.552	23.4 ± 3.7	22.3 ± 3.7	0.067
High grade hematological malignancies	59 (62.8%)	103 (86.6%)	<0.001	37 (61.7%)	125 (81.7%)	0.002
Disease status			0.278			0.001
Non-progressive (CR, PR, SD)	23 (24.5%)	23 (19.3%)		22 (36.7%) ^a^	24 (15.7%) ^a^	
Relapsed or refractory	26 (27.7%)	45 (37.8%)		11 (18.3%) ^a^	60 (39.2%) ^a^	
Non-evaluable	45 (47.9%)	51 (42.9%)		27 (45%)	69 (45.1%)	
Allogeneic HSCT	15 (16%)	27 (22.7%)	0.22	7 (11.7%)	35 (22.9%)	0.064
GVHD	8 (8.5%)	19 (16%)	0.104	5 (8.3%)	22 (14.4%)	0.233
Prolonged severe neutropenia	20 (21.3%)	40 (33.6%)	0.047	12 (20%)	48 (31.4%)	0.097
Albumin at ICU admission (g/dL)	2.9 (2.5–3.3)	2.6 (2.2–3.0)	0.003	2.9 (2.4–3.3)	2.7 (2.2–3.1)	0.076
Invasive mechanical ventilation	81 (86.2%)	116 (97.5%)	0.002	51 (85%)	146 (95.4%)	0.009
Duration of mechanical ventilation (days)	7 (5–17.3)	8 (3–16.5)	0.202	6 (4–14)	9 (4–18)	0.144
Renal replacement therapy initiation in ICU	13 (13.8%)	62 (52.1%)	<0.001	5 (8.3%)	70 (45.8%)	<0.001
ICU acquired infection ^b^	28 (29.8%)	54 (45.4%)	0.02	15 (25%)	67 (43.8%)	0.011
SOFA score	10 (7.8–12)	13 (10–16)	<0.001	10 (7–11.8)	12 (10–16)	<0.001
SAPS II	56.5 (48–66)	73 (58.5–85)	<0.001	53 (45.3–64.8)	67 (56–82)	<0.001
APACHE II score	26 (20.8–30)	31 (25–36)	<0.001	24 (19–29)	29 (24–35)	<0.001

^a^: The adjusted standardized residual was greater than 2 which indicates the column proportions were significantly different at *p* < 0.05 level; ^b^*:* Positive culture results >48 h after ICU admission. Abbreviations: ICU = intensive care unit; CR = complete remission; PR = partial remission; SD = stable disease; PD = disease progression; HSCT = hematopoietic stem cell transplantation; GVHD = graft versus host disease; SOFA = Sequential Organ Failure Assessment; SAPS = Simplified Acute Physiology Score; APACHE = Acute Physiology and Chronic Health Evaluation.

**Table 3 jcm-12-00958-t003:** Univariate analysis and multivariate logistic regression analysis of the prognostic variables for ICU mortality.

Variables	Univariate Analysis with Logistic Regression Analysis			Multivariate Analysis with Logistic Regression Analysis ^a^		
	OR	95% CI	*p* Value	OR	95% CI	*p* Value
High-grade hematological malignancies	3.82	1.95–7.48	<0.001	2.84	1.24–6.50	0.013
Disease status						
Non-progressive (CR, PR, SD)	Ref	--	--	Ref	--	--
Relapsed or refractory	1.73	0.82–3.68	0.153	1.74	0.67–4.52	0.254
Non-evaluable	1.13	0.56–2.29	0.727	0.79	0.32–1.92	0.595
Allogeneic HSCT	1.55	0.77–3.11	0.222			
Prolonged severe neutropenia	1.87	1.00–3.49	0.048	1.25	0.57–2.75	0.584
Albumin at ICU admission (g/dL)	0.48	0.29–0.77	0.003	0.57	0.32–1.01	0.055
Invasive mechanical ventilation	6.21	1.71–22.48	0.005	7.69	1.59–37.09	0.011
Duration of mechanical ventilation (days)	0.99	0.97–1.01	0.198			
Renal replacement therapy initiation in ICU	6.78	3.41–13.48	<0.001	5.61	2.57–12.24	<0.001
SOFA score	1.24	1.15–1.35	<0.001	1.16	1.06–1.28	0.002

^a^: Hosmer and Lemeshow test: *p* = 0.925; Abbreviations: ICU = intensive care unit; CR = complete remission; PR = partial remission; SD = stable disease; PD = disease progression; HSCT = hematopoietic stem cell transplantation; SOFA = Sequential Organ Failure Assessment; SAPS = Simplified Acute Physiology Score; APACHE = Acute Physiology and Chronic Health Evaluation.

**Table 4 jcm-12-00958-t004:** Multivariate logistic regression analysis of the prognostic variables for hospital mortality.

Variables	Multivariate Analysis with Logistic Regression Analysis ^a^			
	β-Coefficient	OR	95% CI	*p* Value
High grade hematological malignancies		1.99	0.84–4.70	0.117
Disease status				
Non-progressive (CR, PR, SD)		Ref.	--	--
Relapsed or refractory	1.849	6.35	2.21–18.29	0.001
Non-evaluable	1.023	2.78	1.05–7.36	0.039
Allogenic HSCT		2.62	0.46–14.89	0.277
GVHD		1.02	0.14–7.67	0.981
Prolonged severe neutropenia		0.96	0.39–2.41	0.938
Invasive mechanical ventilation	1.385	4.00	1.03–15.50	0.045
Renal replacement therapy initiation in ICU	1.794	6.01	2.09–17.28	0.001
SOFA score ≥ 11	0.98	2.66	1.23–5.76	0.013
ICU-acquired infection	0.883	2.42	1.08–5.43	0.032

^a^: Hosmer and Lemeshow test: *p* = 0.225. Abbreviations: CR = complete remission; PR = partial remission; SD = stable disease; PD = disease progression; HSCT = hematopoietic stem cell transplantation; GVHD = graft versus host disease; ICU = intensive care unit; SOFA = Sequential Organ Failure Assessment.

**Table 5 jcm-12-00958-t005:** HHM score for the prediction of hospital outcome in patients with hematological malignancies admitted to the intensive care unit.

Item	Point
Relapsed or refractory disease	2
Non-evaluable disease status	1
Invasive mechanical ventilation	2
Renal replacement therapy initiation in ICU	2
SOFA score ≥ 11	1
ICU acquired infection	1
Total	9
Score	Prediction of hospital mortality
0–1	≤20%
≥3	>48%
≥5	>80%
≥7	100%

## Data Availability

The datasets used and analyzed during the current study are available from the corresponding author on reasonable request.

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
