# Peer review of "Outcomes and Prognostic Factors in Critical Patients with Hematologic Malignancies"

_jcm, 2023, doi:10.3390/jcm12030958_

Round 1

Reviewer 1 Report

Reviewer Comment:

The study is interesting and answered to the objective of research that aims to determine the ICU and hospital outcomes and risk factors that predict the prognosis of critically ill patients with HMs.

However, please clarify following:

 1.  Please consider to clarify the ICU and hospital mortality are all-cause of mortality.

2. Table 3, Please add about how authors select variables with a p-value <0.05 in univariable analysis for the multivariable model. It seems to be more common to use p <0.2 for the selection.

3. Please add the reason to selected the cut point of SOFA ≥11 to develop scoring because authors select SOFA as continuous data in multivariable analysis.

Best regards,

Reviewer 2 Report

Chen and colleagues investigated in this study outcme of critically ill patients with HHM and identified prognostic factors.

Several authors have previously reported similar studies. However, this  study presents some originality due to the population and the study period. They found high ICU and in-hospital mortality and that severity of organ dysfunction was associated with ICU mortality and disease status with in-hospital mortality.

The reader have some comments and criticisms

1- It is important to the reader to investigate the impact of performance status on outcome. The authors should investigate this parameter.

2- What was the policy in this hospital regarding to admission of patients with HHM to the ICU? Which relation with hematologists?

3- The authors classified some hematological diseases as high grade. It seems to the reader that status of the disease is more important. The authors should discuss this

4- Were non evaluable HHM correspond to unexpected admissions to ICU?

5- The authors should define and describe infections acquired during ICU stay. Did the authors have any data about these infections?

6- It has been shown that the evolution of SOFA score during ICU stay is more pertinent that SOFA at 24hours of ICU. Did the authors have the data of SOFA at day 3 or 5 of ICU stay?

7- The authors reported the number of patients invasively ventilated. However, it is interesting to report the impact of duration of mechanical ventilation. This comment will be also for renal replacement therapy.

8- We do not have any data about vasoconstrictive agents since the authors some patients admitted for septic shock.

9- The authors reported reasons of ICU admission only for 183/213 patients. The authors should justify so

10- Some patients were admitted for hematologic emergencies and other for neurological disorders. Please clarify that.

11- In Table 1, GVH should be classified with allogenic HSM transplantation.

12- What was cause of death in this population? how many paatients deceaded after implementation of end-of-life?

13- As pointed by the authors, the score should be validated in another cohort.

14- in discussion section, the authors overstated the role of sofa score in triage.   

Reviewer 3 Report

Studies on prognosis indicators and mortality markers in patients with hematological malignancies guide clinicians in patient follow-up. There are many studies on this subject, but the HHM score evaluation adds originality to the study. The data of the study are detailed, and statistical analysis is appropriate. The results are nicely explored and discussed in detail. However, the material and method part can be written in more detail. Here are a few suggestions for work: 1. If the SOFA score was to be selected for further analysis, was SAPS II and APACHE II score analysis necessary? Today, the SOFA score is mainly used in intensive care units. If the prognostic value of SAPS II, APACHE II, and SOFA score is to be compared, this is the subject of another study. It can cause confusion in the data. 2. Definitions such as intensive care mortality and hospital mortality should be made in more detail in the material and method section. 3. Why was the value 11 chosen for the Sofa score? Is there any scientific proof? It should be explained in detail in the text.

The supplementary table contains data from previous studies found in the literature. It may be more appropriate to include the table in the discussion section.

Round 2

Reviewer 2 Report

The reader find responses adequate and manuscript improved.